# Identification of Novel Genomic Associations and Gene Candidates for Grain Starch Content in Sorghum

**DOI:** 10.3390/genes11121448

**Published:** 2020-12-02

**Authors:** Sirjan Sapkota, J. Lucas Boatwright, Kathleen Jordan, Richard Boyles, Stephen Kresovich

**Affiliations:** 1Advanced Plant Technology Program, Clemson University, Clemson, SC 29634, USA; jboatw2@clemson.edu (J.L.B.); kjorda7@clemson.edu (K.J.); skresov@clemson.edu (S.K.); 2Department of Plant and Environmental Sciences, Clemson University, Clemson, SC 29634, USA; rboyles@clemson.edu; 3Pee Dee Research and Education Center, Clemson University, Florence, SC 29506, USA

**Keywords:** grain composition, cereal crops, association mapping, starch, sorghum, gene network, genomics, mixed-linear models

## Abstract

Starch accumulated in the endosperm of cereal grains as reserve energy for germination serves as a staple in human and animal nutrition. Unraveling genetic control for starch metabolism is important for breeding grains with high starch content. In this study, we used a sorghum association panel with 389 individuals and 141,557 single nucleotide polymorphisms (SNPs) to fit linear mixed models (LMM) for identifying genomic regions and potential candidate genes associated with starch content. Three associated genomic regions, one in chromosome (chr) 1 and two novel associations in chr-8, were identified using combination of LMM and Bayesian sparse LMM. All significant SNPs were located within protein coding genes, with SNPs ∼ 52 Mb of chr-8 encoding a Casperian strip membrane protein (CASP)-like protein (*Sobic.008G111500*) and a heat shock protein (HSP) 90 (*Sobic.008G111600*) that were highly expressed in reproductive tissues including within the embryo and endosperm. The HSP90 is a potential hub gene with gene network of 75 high-confidence first interactors that is enriched for five biochemical pathways including protein processing. The first interactors of HSP90 also showed high transcript abundance in reproductive tissues. The candidates of this study are likely involved in intricate metabolic pathways and represent candidate gene targets for source-sink activities and drought and heat stress tolerance during grain filling.

## 1. Introduction

Carbon is the currency of trade in metabolic engineering of photosynthates in plants. In this carbon trade, the seeds of cereal crops serve as an ultimate sink where sugars are translocated for formation of endosperm that serves as reserve energy during germination. This carbon allocation in the endosperm of cereal grains occurs primarily in the form of starch, which is one of the most important sources of human and animal nutrition. Sorghum (*Sorghum bicolor*. (L.) Moench) is a cereal crop that provides a dietary staple for over half a billion people in semiarid tropics [1]. While primarily used as animal feed in industrialized economies, the end-use products of sorghum grain has diversified to include baking, malting, brewing, and bio-fortification [2]. Most of these end-use products are based on starch which constitutes 60–80% of the dry weight of sorghum grain [3]. Furthermore, the growing demand for gluten-free foods and beverages for people with coeliac disease and other intolerances to gluten in wheat and barley has fueled the demand for gluten-free sorghum grain [4]. As such, understanding the genetic basis of phenotypic variation in grain starch content is imperative for metabolic engineering of this macromolecule through selective breeding for various end-use products.

Starch is composed of two glucose homopolymers: amylose and amylopectin [5]. During starch synthesis, glucose-1 phosphate is converted to ADP-glucose, which is then used by the starch synthase enzyme via a 1,4-alpha glycosidic bond to form a linear chain of glucose residues that becomes amylose after ADP is liberated. Subsequently, the starch branching enzyme introduces 1,6-alpha glycosidic bonds between the amylose chains to form branched amylopectin [6]. Starch is an integrator of plant growth and hence can serve as either a source or sink depending on developmental stage, diurnal cycle, tissue type, and environmental condition [7]. Starch stored in chloroplasts of leaves is used as an energy source for cellular respiration at night, whereas starch that is translocated into the storage organs serves as sink [8]. Starch in cereal grains is stored primarily in the endosperm, which is an important tissue from multiple points of view: (1) plant breeding—because it is a major component of yield—and (2) evolutionary biology and physiology—because it provides energy required for germination and early growth [9]. A positive correlation between grain yield and grain starch content has also been reported in cereals including sorghum [10,11]. While the starch synthesis pathway is well defined in maize, the lack of diversity and strong selection at those loci hint at the necessity for allele mining and characterization of wider diversity in maize and related species [12]. The major genes involved in starch biosynthesis and metabolism are conserved between sorghum and maize [11]. However, only the granule bound starch synthase (GBSS) enzyme (commonly referred to as waxy) that controls amylose content in grain has actually been completely characterized in sorghum [13].

Linkage mapping has been a powerful method for identifying quantitative trait loci (QTL) that co-segregate with a given trait but suffers from two fundamental limitations; only allelic diversity that segregates between the parents can be assayed, and the amount of recombination from bi-parental crosses places a limit on the mapping resolution [14]. In contrast, genome-wide association studies (GWAS) have mapped genetic variants associated with phenotypes to a much higher resolution using whole-genome markers in a diverse group of individuals. However, in association mapping, the complex patterns of genetic relatedness among individuals arising from non-random mating can be problematic for traits that are correlated with population structure [15]. Initially, the ancestry (Q) matrix or eigen vectors from principal component analysis (PCA) calculated using marker data were used to control for population structure in association mapping, but then, application of mixed model methods using the kinship (K) matrix demonstrated that correction for pairwise relatedness between individuals significantly decreases false positives and false negatives compared to corrections involving only the Q matrix [16,17,18]. Subsequently, mixed models using the combined (**Q** + **K**) approach showed increased statistical power to control for false positives in many cases [15]. Several software programs have been developed to run GWAS on plant and animal populations with Genome Association and Prediction Integrated Tool (GAPIT) [19] and Genome-wide efficient mixed model association (GEMMA) [20] being the two most commonly used platforms among plant breeders. GEMMA (Genome-wide Efficient Mixed Model Association) is open sourced and computationally efficient for fitting standard linear mixed models and its relatives for large-scale GWAS. The platform fits both univariate and multivariate linear mixed models while controlling for population stratification and genetic relatedness between individuals [20,21]. GEMMA also fits a Bayesian sparse linear mixed model (BSLMM) using Markov chain Monte Carlo (MCMC) to estimate phenotypic variance explained (PVE) by typed genotypes and to identify associated markers by jointly modeling all markers while controlling for population structure [22].

Previous association analyses for starch content have identified some significantly associated genomic regions in sorghum [3,11,23]. However, mapping the genetic variants associated with grain starch content have only explained a small proportion of the phenotypic variance. A long-term selection experiment in maize suggested that kernel starch content is quantitatively inherited and controlled by many small effects loci [24]. Therefore, approaches that combine different models—including those that can jointly account for effects of variants across multiple loci—could be an effective strategy. Furthermore, the path from genetic association to biological basis is not always straightforward because an association between a genetic variant and a trait may not be informative with respect to the target gene [25]. In this study, we implemented a linear mixed model and a sparse linear mixed model for association mapping of starch content using sorghum association panel and identified candidate genes within significantly associated loci. We also performed candidate gene network analysis using protein–protein interaction data and studied baseline expression profiles of candidate genes and their interactors.

## 2. Materials and Methods

### 2.1. Plant Material

A panel of approximately 400 diverse sorghum accessions was planted in randomized complete block design with two replications in 2013, 2014, and 2017 field seasons at the Clemson University Pee Dee Research and Education Center in Florence, SC. This diversity panel, with over 80% of the accessions from the original United States sorghum association panel (SAP) developed by Casa et al. [26], will be referred to as SAP. Complete details on experimental field design and agronomic practices have been described in details in Boyles et al. [27] and Sapkota et al. [28]. Succinctly, the experiments were planted in two-row plots each 6.1 m long, separated by row spacing of 0.762 m with an approximate density of 130,000 plants ha${}^{-1}$. Fields were irrigated only when signs of drought stress were seen across the field. Primary panicle of three plants selected from each plot was harvested at physiological maturity. The plants from beginning and end of the row were excluded to account for border effect. Panicles were air dried to a constant moisture (10–12%) and threshed; 25 g of a cleaned and homogenized subsample of grain ground to 1-mm particle size with a CT 193 Cyclotec Sample Mill (FOSS North America) was used in near infrared spectroscopy (NIRS) for compositional analysis.

### 2.2. Phenotypic Data

A DA 7250™ NIR spectrometer (Perten Instruments) was used for compositional analysis. The predicted phenotypic values for starch content (% dry matter basis) were obtained from the calibrated curves for spectral measurements of ground grain samples. The calibration curve was built using wet chemistry values from a subset of samples. The wet chemistry was performed by Dairyland Laboratories, Inc. (Arcadia, WI, USA) and the Quality Assurance Laboratory at Murphy-Brown, LLC (Warsaw, NC, USA). The details on the prediction curves and wet chemistry can be found in Boyles et al. [11].

In the first step, we performed phenotypic data analysis to adjust for environmental effects and to calculate variance components. The phenotypic values were evaluated with a linear mixed model analysis using the lme4 package in R [29,30]. The following mixed model equation was fit:(1)$$y_{ijk}∼G_{i}+Y_{j}+G_{i}\times Y_{j}+Y_{j}\times R_{k}+\epsilon_{ijk},$$
where $y_{ijk}$ represents the phenotypic value for the combination of genotype *i*, year *j*, and replication *k*; $G_{i}$, $Y_{j}$, $G_{i}\times Y_{j}$, and $Y_{j}\times R_{k}$ are random effects of genotype, year, genotype-by-year, and replication nested within the year, respectively; and $\epsilon_{ijk}$ is the random effect of residuals, with $N(0,\sigma_{\epsilon }^{2})$. The model in Equation (Equation 1) only took into consideration the genotypic (line) information and not genomic marker data for the genotypes. The best linear unbiased predictors (BLUPs) of random genetic effects for genotypes ($G_{i}$) from the linear mixed model were calculated and used as adjusted phenotypic values in subsequent association analysis. Variance components were used to calculate the broad sense heritability:(2)$$H^{2}=\frac{\sigma_{G}^{2}}{\sigma_{G}^{2}+\frac{\sigma_{G\times Y}^{2}}{Y}+\frac{\sigma_{\epsilon }^{2}}{YR}}.$$

### 2.3. Genotypic Data

The population was genetically characterized using genotyping-by-sequencing [27,31]. Sequenced reads were aligned to the BTx623 v3.1 reference assembly (phytozome) using Burrows-Wheeler aligner [32]. TASSEL 5.0 pipeline was used for SNP calling, imputation, and filtering [33]. The missing genotypes were imputed using the TASSEL plugin FILLIN [34]. Following imputation SNPs with minor allele frequency (MAF) <0.01, sites missing in more than a third of the genotypes in diversity panel were filtered. Genotypes with more than 10% of SNP sites missing were filtered. The SNP genotype file was converted into plink [35] binary ped and bed format for association and linkage disequilibrium (LD) analysis. LD between associated SNPs and neighboring SNPs were calculated using PLINK [35].

### 2.4. Genome-Wide Association Analysis

In the second step, genome-wide association analysis (GWAS) between SNPs and adjusted phenotypic values (BLUPs) was conducted using two models implemented using the software GEMMA v0.94 [20,21]. Only the SNPs with minor allele frequency (*-maf*) of at least 5% and missingness (*-miss*) of less than 33% (present in at least two-third of the genotypes) were used for association analysis.

#### 2.4.1. Linear Mixed Model

First, a **Q** + **K** univariate linear mixed model (LMM) was fit as given by the following equation:(3)$$y=W\alpha +x\beta +u+\epsilon ;u∼{MVN}_{n}\left(0,\lambda \tau^{-1}K\right),\epsilon ∼MVN_{n}\left(0,\tau^{-1}I_{n}\right),$$
where *y* is a vector of BLUPs for *n* individuals where *n* = 389; **W** = ($w_{1}$, $w_{2}$, …, $w_{6}$) is a *n* × *c* matrix of covariates (fixed effects) including a column of 1s and five columns for **Q** (ancestry coefficients); **α** is a vector of corresponding coefficients including the intercept; **x** is a *n*-vector of marker genotypes; β is the effect size of the marker; **u** is an *n*-vector of random effects; **ϵ** is an *n*-vector of errors; $\tau^{-1}$ is the variance of residual errors; λ is the ratio between two variance components; **K** is a known relatedness matrix; and $I_{n}$ is an *n* × *n* identity matrix. $MVN_{n}$ denotes the *n*-dimensional multivariate normal distribution. The standard genomic relationship matrix (K) calculated using the paramater −*gk 2* in GEMMA was used to account for relatedness between individuals. The **Q** matrix (k = 5) previously reported in Sapkota et al. [28] was used as a covariate to control for population structure. The alternative hypothesis $H_{1}:\beta \neq 0$ was tested against the null hypothesis $H_{0}:\beta =0$ for each SNP using Wald’s statistics (−lmm 1) and corresponding *p*-values were calculated.

#### 2.4.2. Bayesian Sparse Linear Mixed Model

Bayesian sparse linear mixed model (BSLMM) is a hybrid model that combines the sparse regression and linear mixed model with a Markov Chain Monte Carlo (MCMC) algorithm for posterior inference. BSLMM fits the equation:(4)$$y=1_{n}\mu +x\beta +u+\epsilon ;u∼MVN_{n}\left(0,\sigma_{b}^{2}\tau^{-1}K\right),\epsilon ∼MVN_{n}\left(0,\tau^{-1}I_{n}\right),$$
where *y* is a vector of BLUPs for *n* individuals where *n* = 389, $1_{n}$ is an *n*-vector of 1s, μ is a scalar representing the phenotype mean, **X** is an n×p matrix of genotypes measured on *n* individuals at *p* genetic markers, β is the corresponding *p*-vector of the genetic marker effects, and other parameters are the same as defined in Equation (Equation 3). The SNP effect size in BSLMM can be decomposed into two parts: α that captures the small effects that all SNPs have, and β that captures the additional effects of some large effect SNPs. In this case, **u** = **Xα** can be viewed as the combined effect of all small effects, and the total effect size for a SNP *i* is given as $\alpha_{i}+\beta_{i}$.

Results from 10 separate runs using 20 × $10^{6}$ sampling steps and 5 × $10^{6}$ burn-in iterations were averaged across the trait to obtain the posterior inclusion probability. A posterior inclusion probability (PIP) > 0.03 (*p*
$<10^{-4}$) was considered significant based on the null distribution of posterior inclusion probability values from 10 simulated data sets (Appendix A).

### 2.5. Gene Expression and Network Analysis

We conducted gene expression, gene interaction, and network analysis on the associated variants and regions in our post-GWAS analysis. Candidate genes from the significantly associated regions were identified using Genome Feature Format (GFF) files annotated for BTx623 v3.1.1 from Phytozome (www.phytozome.net). Custom python codes were used to isolate associated candidate genes from annotation file and to convert gene names from *Sobic* to *Sb* gene format. Once converted, candidate genes from the associated region were used to identify their high-confidence (with protein–protein interaction score ≥ 700) first interactors using sorghum protein interaction data from STRING v11.0 (www.string-db.org).

Gene expression data of reference line BTx623 was obtained from Davidson et al. [36] to examine the baseline gene expression pattern of genes and interactors for nine different tissue types. The tissues included immature leaves, pre- and post-emergence flowers, anther, pistil, whole seed at 5 Days after pollination (DAP), whole seed at 10 DAP, embryo at 25 DAP, and endosperm at 25 DAP. The tissues were harvested and immediately frozen in liquid nitrogren, and total RNA was isolated for construction of RNA-seq libraries. The details on library processing and transcriptome analysis can be found in Davidson et al. [36]. The expression data used in our analysis was obtained from Esemble plant expression atlas (www.ebi.ac.uk).

We only considered epistasis in our post-GWAS analysis. Epistasis between any two variants from different loci was calculated using the PLINK software ([35]). The flag *--epistasis* with the options *--set-by-set* was used to conduct an epistasis test between one set of SNPs with another set of SNPs. For a given quantitative trait, the epistasis analysis in plink fits a linear regression model:(5)$$y=\beta_{0}+\beta_{1}g_{A}+\beta_{2}g_{B}+\beta_{3}g_{A}g_{B},$$
for each inspected variant pair (*A*, *B*), where $g_{A}$ and $g_{B}$ are allele counts; then, the $\beta_{3}$ coefficients were tested for significance.

## 3. Results

### 3.1. Phenotypic Analysis

Phenotypic data included 389 accessions that were grown in randomized complete block design in a single location across three years with two replications per year. We fit a linear mixed model to account for random effects due to year and genotype × year for starch. Year alone did not have any effect on starch content, whereas the genotype × year effect accounted for 14% of the phenotypic variance (Appendix A). Starch was highly heritable with a broad sense heritability ($H^{2}$) of 0.8 (Appendix A). Using the model described by Equation (Equation 1), we obtained the BLUPs for the random genetic effects.

### 3.2. Genome-Wide Association

We fit a univariate linear mixed model using the software GEMMA [20] with kinship (K) matrix and ancestry (Q) matrix to control for population structure and obtained Wald’s statistics (*p*-values) for marker-trait associations. While peaks were observed across several regions, only one SNP (S1_4067535) at ∼4 Mb of chromosome (chr) 1 showed significant association above the Bonferroni significance (α=0.05) threshold (Figure 1). We ran a Bayesian sparse linear mixed model (BSLMM), which implements a multilocus approach with combination of small and large effects for the SNPs. The objective was to reduce incidences of false positives and false negatives by comparison of the two models. The resultant posterior inclusion probability (PIP) for each SNP from BSLMM was superimposed onto corresponding *p*-values of the SNP from LMM (Figure 1). The peak in chr-1 with significant SNPs also had the highest PIP in the BSLMM. However, two regions in chr-8 had the second and third highest PIPs in the BSLMM that overlapped with visible peaks, but *p*-values from LMM fell below the Bonferroni significance threshold. Therefore, we identified three significantly associated genomic regions based on SNPs with BSLMM (PIP > 0.03) that clearly possessed overlapping peaks with the LMM results (Table 1).

### 3.3. Associated Genes

Potential candidate genes from significantly associated SNPs were identified based on the extent of linkage disequilibrium (LD) between SNPs in the associated regions (Appendix A, Table 2). Several SNPs from chr-1 peak had non-synonymous substitutions within the coding region of a locus (*Sobic.001G054500*) for an uncharacterized protein (Table 1). The associated region near 59 Mb of chr-8 had two lipid transfer protein paralogs: *Sobic.008G158332* and *Sobic.008G158400*. The association peak near 52 Mb of chr-8, which had six SNPs all in strong LD, encompassed two genes, *Sobic.008G111500* and *Sobic.008G111600*, within the LD block (Figure 2). The variant S8_51715166 located within coding region of a CASP-like protein 8 (*Sobic.008G111500*) led to a non-synonymous substitution in the gene, whereas the SNPs S8_51719704 and S8_51726098 within the coding region of a heat shock protein 90 (HSP90-6; *Sobic.008G111600*) only led to synonymous substitutions. However, the three intronic variants within the HSP90-6 gene had a modifier effect in the gene (Table 1).

### 3.4. Candidate Gene Expression

We examined the baseline gene expression of associated genes across nine distinct tissues including vegetative stage, flowering, and seed development of BTx623 using publicly available transcriptomic data [36]. BTx623 is the grain sorghum reference line that ranked 14th for total starch content with higher average starch content (71%) than the population mean (68%) across our association panel. The gene expression for candidate genes from chr-1 peak and ∼59 Mb peak of chr-8 were below threshold (0.5 TPM) and missing across the majority of tissue types. Thus, we compared the gene expression profile for the two genes from ∼52 Mb of chr-8 along with two other genes in the neighborhood of HSP90-6 gene (Figure 3). The two neighboring genes were identified based on neighborhood conservation for the HSP90-6 gene using gene tree viewer in Gramene (Appendix A). The two genes in the neighborhood of HSP90-6 were sugar/carbohydrate transporter family proteins *Sobic.008G111300* and *Sobic.008G111100* located ∼60 Kb and ∼150 Kb upstream of the HSP90-6 gene, respectively (Appendix A). Among the four genes from ∼52 Mb of chr-8, all but *Sobic.008G111300* were poorly expressed in the leaf tissue compared to most of the reproductive tissues (Figure 3). The CASP-like protein (*Sobic.008G111500*) and one of the transporter proteins (*Sobic.008G111300*) showed lower transcript abundance in inflorescence tissues and early grain filling, whereas the HSP90-6 (*Sobic.008G111600*) gene showed higher expression in earlier reproductive stages. CASP-like protein had the highest abundance in the endosperm followed by the embryo, and transcript abundance increased in 10 days after pollination (DAP) compared to 5 DAP. In contrast, the heat shock protein 90-6 ((*Sobic.008G111600*)) showed lower transcript abundance in the endosperm but had the highest abundance in the embryo, and expression decreased at 10 DAP compared to 5 DAP.

### 3.5. HSP90-6 Gene Network Analysis

Heat shock proteins are known to be molecular chaperones primarily involved in drought and stress response but can also be involved in other molecular processes during plant development [37,38]. There are a total of seven paralogs of HSP90 in sorghum. Baseline gene expression of all seven paralogs show that they are highly expressed in early reproductive tissues and the embryo but are less prevalent in leaf, endosperm, and seeds 10 DAP (Appendix A).

We searched for high-confidence (protein–protein interaction (PPI) ≥ 700) first interactors of the HSP90-6 (*Sobic.008G111600*) gene and identified a total of 75 genes that interacted with HSP90-6. The gene interaction network was significantly enriched (False discovery rate (FDR) < 0.001) for five biochemical pathways and four UniProt keywords (Appendix A). Furthermore, a number of protein domains were also found to be significantly enriched. Two studies involving gene networks for drought tolerance in sorghum were also found to be significantly enriched [39,40]. Four genes in our network were found to be involved in drought-related gene networks in those two studies with each investigation involving separate HSP70 genes from our network. Starch is an important determinant of plant fitness under abiotic stress, and plants with increased tolerance to heat and drought are likely to develop stronger sink strength [41,42]. Figure 4 shows an expression atlas for the first interactors across different reproductive tissues of sorghum. In Figure 4, the hierarchical clustering of the tissue types along the column separates the highly expressed early reproductive tissues from the leaf, seeds, and endosperm. This observation is similar to the expression pattern of the heat shock protein that these genes interact with. The row clustering in Figure 4 shows four distinct clusters of genes ranging in different levels of transcript abundance.

To follow up on gene interactions, we took the SNPs from within the genes in the network and tested for any significant epistatic interaction among SNPs from the ∼52 Mb peak of chr-8. In the set-by-set analysis, one set represented SNPs from ∼52 Mb peak of chr-8 and the other set consisted of SNPs from genes that showed high protein–protein interaction with the candidate gene heat shock protein 90-6 (*Sobic.008G111600*). Five SNPs showed significant epistatic interaction with at least five SNPs across five genes in the network (Appendix A). Among the five genes were a HSP70 (*Sobic.001G418600*), two putative peptidyl-prolyl cis-trans isomerase genes (*Sobic.010G085600* and *Sobic.004G306700*) with HSP90 co-chaperone domain, a putative SGT1-1 gene (*Sobic.003G077800*), and a putative uncharacterized protein (*Sobic.004G002200*). The HSP70 (*Sobic.001G418600*) gene was highly expressed in reproductive tissues and was located within 27 Kb of a glycosyltransferase family protein (*Sobic.001G418800*). The glycosyltransferase protein is a branching enzyme responsible for catalysis of the transfer of an N-acetylglucosaminyl residue from UDP-N-acetyl-glucosamine to a sugar.

## 4. Discussion

A large proportion of genetic variance in grain starch content remains unexplained despite starch being one of the most important and frequently studied grain quality traits. The polygenic and quantitative inheritance of starch content complicates the characterization of the genetic controls mediating phenotypic variance [24]. Additionally, the incorporation of the genotype × environment effects have resulted in inconsistent GWAS results, as shown by poor year-to-year SNP correlations [11]. Since we observed a strong genotypic effect and a small genotype × environment effect for starch content in our population, we used a single random genotypic effect (BLUP) as the phenotype with the objective to identify marker trait associations that are purely genetic effects and not confounded by genotype × environment effects.

The strong genotypic effect and high heritability observed in our population is consistent with previous studies for starch in different populations and environments [3,10,11,43]. Boyles et al. [11] (2017) had identified a QTL spanning 1–7 Mb of chr-1 for multiple grain quality traits including starch using sorghum recombinant inbred lines but did not see any significant peak for starch chr-1 in their association mapping result. Additionally, Murray et al. [10] previously identified a starch QTL between 1–5 Mb of chr-1 using a Rio × BTx623 recombinant inbred line (RIL) population, and chr-1 peak in our association analysis lies within the previously identified QTL. The starch QTL in Murray et al. [10] had a stronger correlation with positive sugar yield QTL than the grain yield QTL in that population. Because the starch QTL was colocalized with positive sugar yield QTL, it is likely that our candidate gene (*Sobic.001G054500*) that contained significantly associated variants within non-synonymous coding sequences could be a signaling peptide important for sugar transport during grain filling.

One of the biggest challenges in association analysis is avoiding false positives and false negatives [14,44]. Overconservative approaches such as Bonferroni correction used to avoid type I error could lead to inflation of type 2 errors, especially for quantitatively inherited traits with possible small effect loci [45]. Since BSLMM combines the advantages of both standard LMMs and sparse regression modeling, we decided to combine results from univariate LMM and BSLMM models to avoid false positives and false negatives in our association results. By adopting this approach, we were able to identify two more starch-associated genomic regions that were below the Bonferroni significance threshold in LMM but had high posterior inclusion probability in BSLMM. These two chr-8 peaks are novel associations that have not been identified in previous association mapping studies for starch in sorghum [3,11,23]. One of the associated SNPs, S8_59121722, from the ∼59 Mb chr-8 peak was located within the coding region of the lipid transfer protein, *Sobic.008G158332*, which could play an important role in protein transport during grain filling and therefore requires further analysis. Another peak in chr-8∼52 Mb had five SNPs in strong LD with each other from a single locus that encodes for a heat shock protein (HSP) 90 (*Sobic.008G111600*), and another SNP which was located in the non-synonymous site in the coding region of a CASP-like protein (*Sobic.008G111500*).

The Casperian strip membrane protein (CASP) family is a highly conserved group of plant cell membrane proteins that mediate the deposition of specialized structures, Casparian strips, in the endodermis by recruiting the lignin polymerization machinery [46]. However, they are found to be widely expressed across various organs in arabidopsis and the arabidopsis CASP protein knockout plants have shown altered growth dynamics, faster growth, increased biomass (dry weight), and earlier flowering compared to wild type, suggesting a more fundamental role of CASP-like proteins in vascular tissue [47]. Our candidate CASP-like protein, *Sobic.008G111500*, showed high expression in seed tissues (embryo and endosperm) and increased expression in seeds 10 DAP compared to 5 DAP. Furthermore, the CASP-like protein was located just 4 Kb upstream of a HSP90 gene which showed higher expression in early reproductive tissues, embryo, and seeds 5 DAP but poor expression in seed 10 DAP or the endosperm.

HSPs are a common group of protein found in eukaryotes that function as molecular chaperones and help refold proteins denatured by heat to prevent their accumulation [48,49]. Two distinct members of the Hsp70 family of stress-related protein were localized in the maize amyloplast and form transient complexes with starch synthase 1 (SSI) and other stromal enzymes [37]. In Japanese sake-brewing rice rich in starch content, the HSP70 protein was highly abundant in amyloplast compared to cytosol and its concentration was elevated during the later stages of grain development [50]. Our candidate HSP90 gene showed strong protein–protein interaction with numerous molecular chaperones including several HSP70 proteins. Variants within the coding region of a HSP70 protein, *Sobic.001G418600*, showed significant epistatic interaction with variants within the HSP90-6 gene. The HSP70 protein (*Sobic.001G418600*) had relatively higher expression in seed tissues compared to early reproductive and leaf tissues. Furthermore, the presence of a glycosyltransferase family protein (*Sobic.001G418800*) within 27 Kb of the HSP70 gene suggests a possible role of these genes in source-sink dynamics during grain filling.

Abiotic stresses are known to affect structure and properties of starch in rice [51], and the source-sink dynamics of starch-sugar interconversion is essential for abiotic stress response in plants [42]. Yang et al. [52] showed that heat stress during grain filling reduced starch accumulation by more than 20% as a result of reduced activities of starch synthesis enzymes. The view of starch as merely an inert long-term storage molecule has been increasingly challenged, and the inseparable role of starch in plant growth and adaptation has revealed surprising plasticity of starch metabolism [41]. The association of starch content with genomic regions encoding abiotic stress-related genes, membrane proteins, and potential signaling proteins highlights a more intricate involvement of these group of genes in pathways related to grain filling in sorghum. The large interactions of the HSP90-6 with many genes across the genome suggest that this gene is likely a hub gene responsible in multiple pathways related to the processing and transportation of proteins and sugars during grain filling in sorghum and warrants further investigation into its role in seed development. The strong associations between grain starch content and grain yield as well as between abiotic stress response and grain yield suggest that the suite of genes identified could regulate total starch content through post-flowering heat and drought tolerance [53].

## 5. Conclusions

This study elucidated previously unidentified genomic regions associated with starch content using a combination of LMM and BSLMM. Some associated SNPs were within non-synonymous sites of coding regions of genes from the associated region. One of the genomic regions (∼52 Mb of chr-8) harbored two genes: (1) a CASP-like protein that was highly expressed in the embryo, endosperm, and seed 10 DA, and (2) a heat shock protein 90 which was highly expressed in early reproductive tissues and the embryo, along with a strong gene interaction network that is enriched for several biochemical pathways. The candidates of this study might be involved in intricate metabolic pathways and might represent candidate gene targets for source-sink activities and abiotic stress tolerance during grain filling.

## Figures and Tables

**Figure 1 genes-11-01448-f001:**
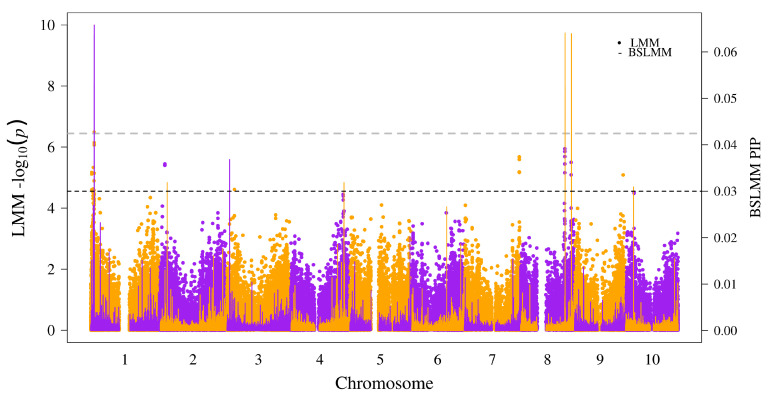
Superimposed Manhattan plot for genome-wide association using a linear mixed model (LMM) and a Bayesian sparse linear mixed model (BSLMM): the chromosome numbers and respective positions of the SNPs are shown on the x-axis. The left y-axis label shows negative $log_{10}$ of *p*-value from Wald’s test for linear mixed model, and the right y-axis label shows the posterior inclusion probability (PIP) for the BSLMM model. The gray dashed line represents the Bonferroni-corrected significance threshold for α = 0.05, and the black dashed line represents a PIP threshold of 0.03.

**Figure 2 genes-11-01448-f002:**
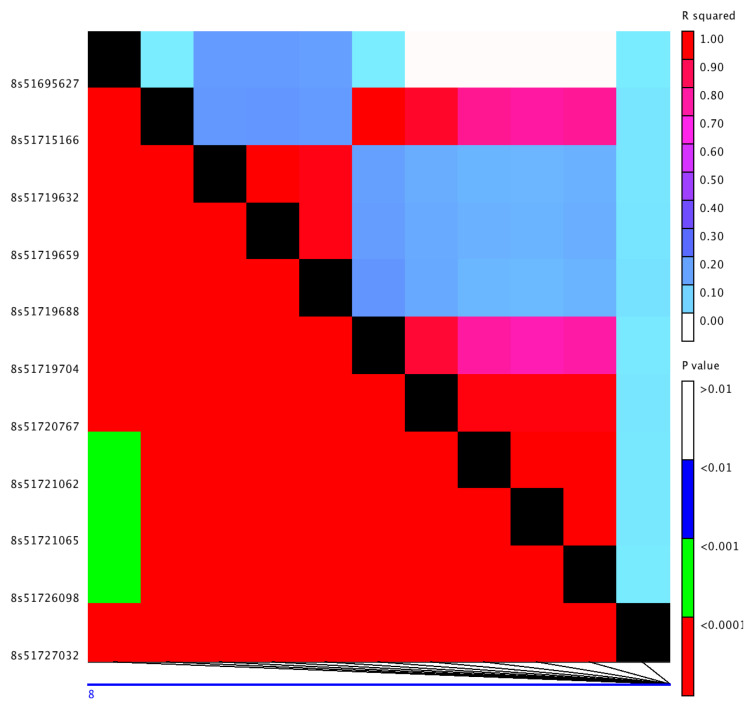
Linkage disequilibrium between single nucleotide polymorphisms (SNPs) from associated region ∼52 Mb of chromosome 8: R-squared values are above the diagonal, and associated *p*-values are below the diagonal. The left y-axis label shows the position of the SNPs with 8s corresponding to Chr8.

**Figure 3 genes-11-01448-f003:**
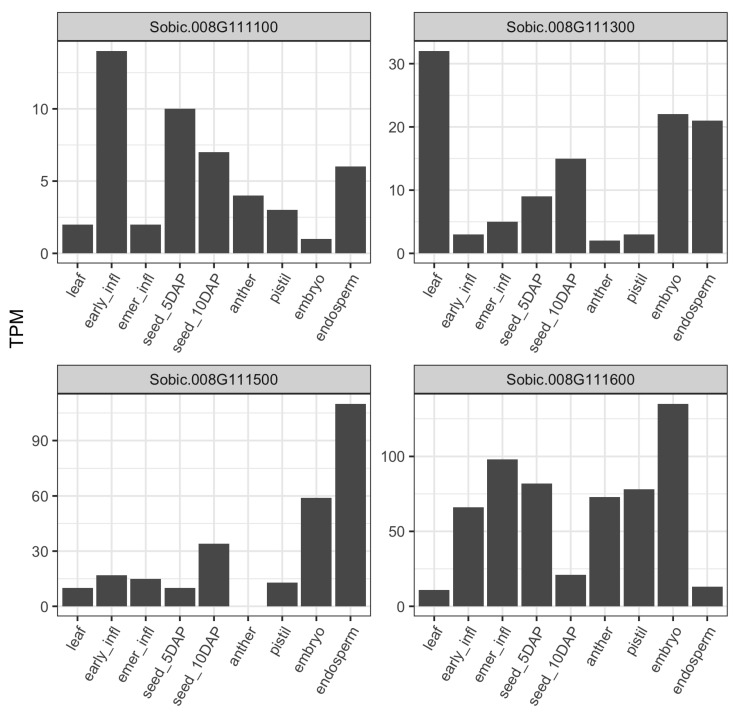
Gene expression profile of genes near 52 Mb associated regions: the Y-axes represent transcript abundance in transcript per million (TPM), and x-axes correspond to tissue type. infl: infloresence, emer: post-emergence, DAP: days after pollination.

**Figure 4 genes-11-01448-f004:**
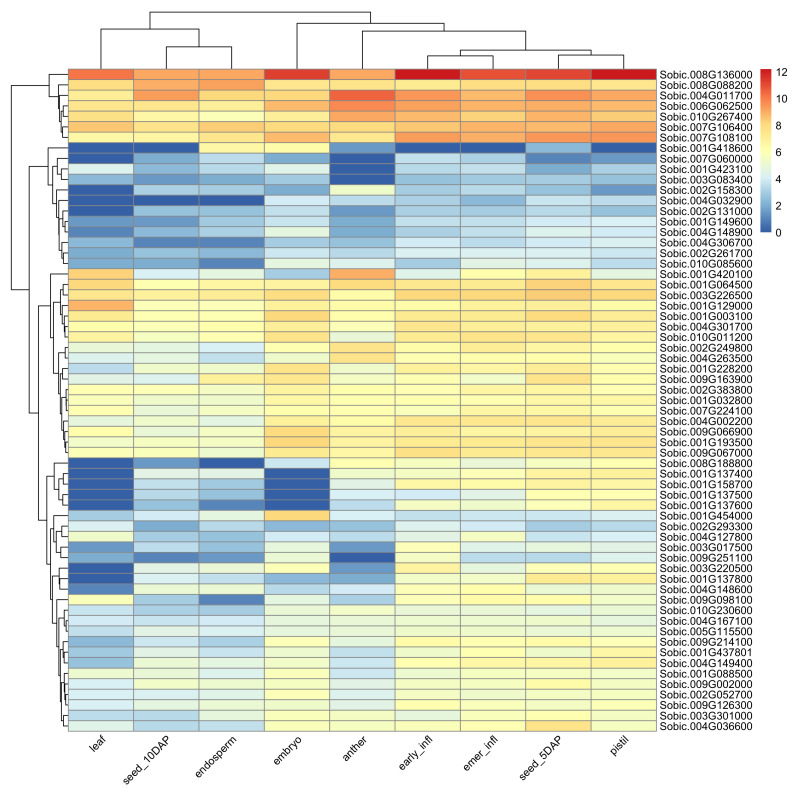
Heatmap showing gene expression analysis of interactors of heat shock 90-6 gene: each row represents a gene, the column corresponds to tissue type, and trees on the left and top show hierarchical clustering of the row and column, respectively. Values in legends correspond to $log_{2}$ of transcript per million (TPM). infl: infloresence, emer: post-emergence, DAP: days after pollination.

**Table 1 genes-11-01448-t001:** Variants from associated regions, their substitution effect, and association statistics. Chr: chromosome, LMM: linear mixed model, BSLMM: Bayesian sparse linear mixed model, and PIP: posterior inclusion probability.

Chr	Position	−log10(p) (LMM)	PIP (BSLMM)	Rank (LMM,BSLMM)	GeneID	Annotation	Impact
1	4067535	6.812	0.066	1,1	Sobic.001G054500	Missense	Moderate
1	4067364	6.412	0.064	2,4	Sobic.001G054500	Missense	Moderate
1	4067377	6.332	0.053	4,5	Sobic.001G054500	Missense	Moderate
8	51715166	5.350	0.022	20,20	Sobic.008G111500	Missense	Moderate
8	51719704	5.682	0.047	12,8	Sobic.008G111600	Synonymous	Low
8	51720767	6.171	0.064	6,2	Sobic.008G111600	Intron	Modifier
8	51721062	6.009	0.053	7,6	Sobic.008G111600	Intron	Modifier
8	51721065	5.728	0.039	10,9	Sobic.008G111600	Intron	Modifier
8	51726098	6.229	0.051	5,7	Sobic.008G111600	Synonymous	Low
8	59121722	5.260	0.064	23,3	Sobic.008G158332	Synonymous	Low

**Table 2 genes-11-01448-t002:** Potential candidate genes from the significantly associated regions.

Gene	Name	Chromosome	Start	End	Maize Homolog
Sobic.001G054500	Uncharacterized protein	1	4066711	4067588	Zm00001d034482
Sobic.008G111500	CASP-like protein 8	8	51714673	51715254	Zm00001d023936
Sobic.008G111600	Heat shock protein 90-6	8	51719209	51726960	Zm00001d041719
Sobic.008G158332	Lipid transfer protein 1	8	59121190	59129810	Zm00001d027290

## Data Availability

The codes and phenotypic data used in the study can be accessed through GITHUB at sirjansapkota/StarchGWAS.

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
