# Peer review of "Identification of Novel Genomic Associations and Gene Candidates for Grain Starch Content in Sorghum"

_genes, 2020, doi:10.3390/genes11121448_

Round 1

Reviewer 1 Report

The manuscript is well structured and written. I have only few minor comments:

L 131: Write „HSP90-6“ not „HSP 90-6“ L

162: Please revise sentence, “…,we extracted the SNPs from within the genes from the network and ...“

L 165: You may write „HSP70“ instead of „heat shock protein 70“, HSP was explained earlier (L 9) and HSP70 already mentioned (L 154).

L 195-201: I am fine with the comments regarding false positives but I am not really convinced regarding false negatives. I could imagine there are still many unidentified loci with an effect on starch content in the population.

Author Response

Thank you for you reviews. The comments will be incorporated into the revised manuscript.

Reviewer 2 Report

Please find comments in the attached file.

Author Response

Dear Reviewer,

Thank you for your comments and suggestions. We have tried our best to address them. Please see the detailed response in the document attached.

Best,
Sirjan Sapkota

Round 2

Reviewer 2 Report

Please find comments in the attached file.

Author Response

Dear reviewer,

Thank you for your suggestions. We hope we have attended to all of your concerns. Please find the detailed response attached.

Thank you,
Sirjan

Round 3

Reviewer 2 Report

This revised version is substantially improved. The questions I had on the previous versions are now cleared, and the M&M is much better and possible to follow and understand. I have carefully revised the manuscript along with the comments made in the previous round, and I believe that this version is almost good for publication, once the minor comment below is addressed.

Minor

L155-156: It is not very clear to me how is it that the BSLMM estimates the \alpha_i and \beta_i to explain the small and large SNP effects.